# Effect of Pregabalin Combined with Duloxetine and Tramadol on Allodynia in Chronic Postischemic Pain and Spinal Nerve Ligation Mouse Models

**DOI:** 10.3390/pharmaceutics14030670

**Published:** 2022-03-18

**Authors:** Jie Quan, Jin Young Lee, Hoon Choi, Young Chan Kim, Sungwon Yang, Jongmin Jeong, Hue Jung Park

**Affiliations:** 1Department of Anesthesiology and Pain Medicine, Seoul St. Mary’s Hospital, College of Medicine, The Catholic University of Korea, Seoul 06591, Korea; qj890701@catholic.ac.kr (J.Q.); hoonie83@naver.com (H.C.); yc0323@gmail.com (Y.C.K.); yangswsj@naver.com (S.Y.); marbinorgia@naver.com (J.J.); 2Department of Anesthesiology and Pain Medicine, Samsung Medical Center, School of Medicine, Sungkyunkwan University, Seoul 06351, Korea; l7035@hanmail.net

**Keywords:** allodynia, duloxetine, glial fibrillary acidic protein, pregabalin, tramadol

## Abstract

Although there are various drugs for Neuropathic pain (NP), the effects of single drugs are often not very satisfactory. The analgesic effects of different combinations of pregabalin, duloxetine, and tramadol or the combination of all three are still unclear. Mixtures of two or three drugs at low and high concentrations (7.5, 10, 15, and 20 mg/kg pregabalin; 7.5, 10, 15, and 30 mg/kg duloxetine; 5 and 10 mg/kg tramadol) were administered to chronic postischemic pain (CPIP) and spinal nerve ligation (SNL) model mice. The effects of these combinations of drugs on mechanical allodynia were investigated. The expression of the glial fibrillary acidic protein (GFAP) in the spinal cord and dorsal root ganglia (DRGs) was measured. The combination of pregabalin, duloxetine, and tramadol significantly alleviated mechanical hyperalgesia in mice with CPIP and SNL. After the administration of this drug combination, the expression of GFAP in the spinal cord and DRGs was lower in the CPIP and SNL model mice than in control mice. This result suggests that the combination of these three drugs may be advantageous for the treatment of NP because it can reduce side effects by preventing the overuse of a single drug class and exert increased analgesic effects via synergism.

## 1. Introduction

Neuropathic pain (NP) is caused by a lesion or disease of the somatosensory nervous system and has an important impact on quality of life [1]. The pathogenesis of NP is complicated. Various studies proposed that pathophysiological mechanisms related to neurogenic inflammation and central sensitization contribute to NP. However, NP may result from the combination of multiple mechanisms. Due to this diversity of mechanisms, NP is difficult to treat, mainly because common analgesics have poor therapeutic efficacy against this disease [2].

Pregabalin is a major anticonvulsant drug. In the past 10 years, this drug was increasingly used for pain relief [3]. It exerts its effects by preventing the release of excitatory neurotransmitters by binding with high affinity to alpha-2delta-1, a voltage-dependent calcium channel subunit expressed in the central nervous system [4]. In addition, it reduces the release of glutamate, norepinephrine, and substance P [5].

Duloxetine is an antidepressant that inhibits the reuptake of serotonin and norepinephrine in the central nervous system, leading to an increase in the levels of these neurotransmitters in the synaptic cleft and thereby inhibiting the descending inhibitory pain pathway [6]. Duloxetine affects the voltage dependence of sodium channels in the central nervous system and peripheral nervous system (PNS) [7].

Tramadol is a mu opioid receptor analgesic with weak central effects [8]. It has a unique mechanism of action and pharmacological effects that are different from those of other opioids. Tramadol has a regulatory effect on a variety of mediators involved in pain signal transduction, such as voltage-gated sodium channels, transient receptor potential vanilloid subtype 1 (TRPV1) channels, glutamate receptors, α2 adrenergic receptors (α2-AR), and adenosine receptors, and affects molecules such as substance P, calcitonin gene-related peptide, prostaglandin E2, and proinflammatory cytokines. Through these molecular effects, tramadol can regulate the hyperexcitability of peripheral and central neurons, thereby alleviating NP [9].

Pregabalin, duloxetine, and tramadol all have a certain ability to relieve NP, but these effects are not obvious when they are used individually [10]. There are few reports on the effects of different combinations of pregabalin, duloxetine, and tramadol. This study evaluated the effects of the combinations of these three drugs in two animal models, namely, a chronic postischemic pain (CPIP) model and a spinal nerve ligation (SNL) model, to identify a drug combination with enhanced efficacy and fewer side effects such as dizziness, drowsiness, and motor weakness for development as a novel therapeutic strategy.

## 2. Materials and Methods

### 2.1. Animals

In this study, adult male C57BL/6 mice (weight: 25–30 g) were divided into groups of 5 mice each, provided free access to food and water, and housed under a 12:12 h light:dark cycle. All animals were allowed to adapt to the environmental conditions for 7 days. The animal research protocol was approved by the Institutional Animal Care and Use Committee of the Catholic University of Korea (CUMC-2019-0248-01). All mouse studies were conducted according to the guidelines and regulations governing the use of these animals. The total number of mice used was 224.

### 2.2. Chronic Postischemic Pain (CPIP) Model

The mice were anesthetized with 1.5% isoflurane and 100% O_2_, and according to the method described by Coderre et al. [11], an O-ring with an inner diameter of 5/64 inches (AS568-004), which matched the size of the mouse hind limbs, was placed on the left upper ankle (just above the medial malleolus) for 3 h. In the sham operation group, precut O-rings of the same size were used to prevent tightening. After 3 h of ischemia, the thin O-ring was pulled out to induce reperfusion, and the mice were allowed to recover from anesthesia.

### 2.3. Spinal Nerve Ligation (SNL) Model

SNL was performed based on the method described by Kim and Chung [12]. The mice were anesthetized with isoflurane, and the left paraspinal muscle was dissected at the level of L4 to S2. Under a magnifying glass, the L5 transverse process was carefully excised, and the L4–L6 spinal nerves were located. The L5 spinal nerve was identified, carefully separated from other adjacent tissues, and then tightly ligated with silk thread. The wound was washed with disinfectant and sutured. The mice in the sham operation group were subjected to the same operation under general anesthesia except for the ligation of the spinal nerve.

### 2.4. Drug Administration

SNL and CPIP model mice were intraperitoneally injected with extended-release pregabalin (Yuhan, Seoul, Korea), duloxetine (Lilly, Indianapolis, IN, USA), tramadol (Yuhan, Seoul, Korea), and normal saline. These drugs were dissolved in 0.9% saline (*n* = 4) and injected intraperitoneally after 15 days after surgery. The groups were as follows: the 10 mg/kg pregabalin + 5 mg/kg tramadol (Pre10 + Tra5), 20 mg/kg pregabalin + 10 mg/kg tramadol (Pre20 + Tra10), 7.5 mg/kg pregabalin + 7.5 mg/kg duloxetine (Pre7.5 + Dul7.5), 15 mg/kg pregabalin + 15 mg/kg duloxetine (Pre15 + Dul15), 10 mg/kg duloxetine + 5 mg/kg tramadol (Dul10 + Tra5), 30 mg/kg duloxetine + 10 mg/kg tramadol (Dul30 + Tra10), 5 mg/kg pregabalin + 5 mg/kg duloxetine + 5 mg/kg tramadol (Pre5 + Dul5 + Tra5), and 10 mg/kg pregabalin + 10 mg/kg duloxetine + 10 mg/kg tramadol (Pre10 + Dul10 + Tra10) groups (*n* = 6 per each group). Mechanical hypersensitivity was assessed 30, 60, 90, 120, 180, 240 min, and 24 h after injection. Changes in mobility and motor function in neuropathic mice were assessed by the rotarod test (Acceler Rotarod for the rat 7750; Ugo Basile, Comerio-Varese, Italy). Neuropathic mice were accustomed to rotating drums, and they were accustomed to being carried around to relieve any stress during the test. The day before the actual test, the mice performed 3 training trials on a rotating drum (10–15 rpm) for 2 days. Mice that were able to stay on the drum for at least 150 s were selected for drug testing. The average of 3 training runs was used as the control operation time. Rotor action time was measured at 30, 60, 90, 120, 180, 240 min, and 24 h after intraperitoneal injection, respectively (*n* = 4–6 per group). Each test was performed 3 times every 5 min, and the mean values were compared in the high dose group [13].

### 2.5. Evaluation of Mechanical Allodynia

Mechanical allodynia was measured every 2 days from before the surgery to 15 days after reperfusion. The mice were placed on a wire mesh floor in an 8 × 8 × 18 cm transparent plastic box. After the mouse was allowed to adapt to the environment for approximately 30 min, a von Frey filament (18011 Semmes–Weinstein filament, Stoelting Co., Wood Dale, IL, USA) was applied perpendicularly to the mid-plantar region of the paw for 3 s until the filament bent, and the response of the mouse was evaluated. Seven filaments with forces from 2.44 to 4.31 (0.04–2.00 g) were used. The simplified up–down method described by Bonin et al. [14]. was used to assess the responses of the mice in 4 trials starting from the trial in which the mouse started showing an avoidance response or stopped showing an avoidance response. The 50% response threshold was measured according to the response pattern (positive response: lifting, shaking, licking, etc.) and the value (in log units) of the last von Frey filament used.

### 2.6. Expression of Glial Fibrillary Acidic Protein (GFAP) in the Spinal Cord and Dorsal Root Ganglion (DRG)

The mice were injected with 20 mg/kg pregabalin + 10 mg/kg tramadol (Pre20 + Tra10), 15 mg/kg pregabalin + 15 mg/kg duloxetine (Pre15 + Dul15), 30 mg/kg duloxetine + 10 mg/kg tramadol (Dul30 + Tra10), 10 mg/kg pregabalin + 10 mg/kg duloxetine + 10 mg/kg tramadol (Pre10 + Dul10 + Tra10), or normal saline (*n* = 4–6 per each group). They were sacrificed 90 min later, and the spinal cord and L4-6 DRGs were collected based on the results of the mechanical allodynia test. The mice were anesthetized and subjected to cardiac perfusion with 50 mL of 4% paraformaldehyde dissolved in 0.1 M phosphate-buffered saline (pH 7.2–7.4). Then, the spinal cord and DRGs of the mice were removed. All the obtained tissues were postfixed and then soaked in a 30% sucrose solution overnight. The spinal cord and DRG tissues were cut into 10 μm-thick sections on a cryostat, washed 3 times with 0.1 M PB (phosphate-buffered solution) for 10 min each, and incubated with 10% normal donkey serum for 1 h at room temperature. The sections were incubated with anti-GFAP antibody (GA-5: sc-58766, Mouse Monoclonal—Santa Cruz Biotechnology) at 4 °C overnight. Then, the tissues were rinsed thoroughly in PB and incubated with Alexa Fluor 488-conjugated donkey anti-mouse secondary antibody (A21202, Thermo Fisher Scientific, Waltham, MA, USA 1:1000) at room temperature for 2 h. After rinsing several times in PB, the nuclei were counterstained with DAPI for 10 min, and the sections were fixed with an antifading mounting medium (Vector Laboratories; Burlingame, CA, USA). A Zeiss LSM 800 confocal microscope (Carl Zeiss Co. Ltd., Oberkochen, Germany) was used for observation and image acquisition. ImageJ (National Institutes of Health )and the Laboratory for Optical and Computational Instrumentation (LOCI, University of Wisconsin) was used to measure the average intensity.

### 2.7. Statistics

The data are expressed as the mean ± standard error of the mean. Statistical analysis was performed using GraphPad Prism 8.0 (GraphPad Software, Inc., San Diego, CA, USA). Repeated measures two-way analysis of variance (ANOVA) was used to analyze the significance of the difference in the 50% von Frey threshold at each time point under different conditions, and then the Bonferroni post hoc test was performed. At each time point, a Student’s *t*-test and the Bonferroni post hoc test were used to compare values between the control group and the treatment group. Two-tailed *p*-values < 0.05 were considered statistically significant. One-way ANOVA was used to compare the expression of GFAP among the CPIP group, SNL group, and treatment group.

## 3. Results

### 3.1. CPIP and SNL Model Mice Show Significant Mechanical Allodynia

CPIP model mice exhibited long-term mechanical allodynia in the ipsilateral hind paws (Interaction: DF = 8, F (8,72) = 15.71, *p* < 0.0001; Row Factor: DF = 8, F (8,72) = 11.42, *p* < 0.0001; Column Factor: DF = 1, F (1,72) = 143.4, *p* < 0.0001; two-way ANOVA followed by Bonferroni’s post hoc test). Ipsilateral mechanical allodynia developed within 4 days after 3 h of ischemia–reperfusion, peaked at 9 days, and lasted for more than 15 days after reperfusion (Figure 1A). Ligation of the L5 spinal nerve caused obvious ipsilateral mechanical allodynia (Interaction: DF = 8, F (8,72) = 6.653, *p* < 0.0001; Row Factor: DF = 8, F (8,72) = 9.977, *p* < 0.0001; Column Factor: DF = 1, F (1,72) = 230.7, *p* < 0.0001; two-way ANOVA followed by Bonferroni’s post hoc test). Ipsilateral mechanical allodynia initially developed on the first day after surgery and peaked on the fifth day after surgery (Figure 1B).

### 3.2. The Combination of the Three Drugs Alleviates Mechanical Allodynia in CPIP Model Mice

Intraperitoneal injection of the drug combinations ameliorated mechanical allodynia in CPIP model mice in a dose-dependent manner (Interaction: DF = 56, F (56,344) = 5.140, *p* < 0.0001; Row Factor: DF = 7, F (7,344) = 120.3, *p* < 0.0001; Column Factor: DF = 8, F (8,344) = 39.99, *p* < 0.0001; two-way ANOVA followed by Bonferroni’s post hoc test). Compared with the control treatment, the six drug treatments showed a stronger analgesic effect at 90 min (Pre10 + Tra5, *p* < 0.05; Pre20 + Tra10, *p* < 0.0001; Pre7.5 + Dul7.5, *p* > 0.05; Pre15 + Dul15, *p* < 0.0001; Dul10 + Tra5, *p* > 0.05, Dul30 + Tra10, *p* < 0.0001; Pre5 + Dul5 + Tra5, *p* < 0.001; Pre10 + Dul10 + Tra10, *p* < 0.0001). The Pre7.5 + Dul7.5 and Dul10 + Tra5 groups exhibited no obvious alleviation of allodynia. The Pre10 + Tra5 injection showed an obvious anti-allodynic effect only at 90 min, whereas the Pre5 + Dul5 + Tra5 injection exhibited an obvious anti-allodynic effect at 90 and 120 min. In the Pre15 + Dul15 and Dul30 + Tra10 groups, mechanical allodynia was significantly alleviated from 60 to 120 min, while in the Pre10 + Dul10 + Tra10 group, mechanical allodynia was significantly ameliorated from 30 to 180 min (Figure 2A). Pre10 + Dul10 + Tra10 injection had the best analgesic effect. 

### 3.3. The Combination of the Three Drugs Alleviates Mechanical Allodynia in SNL Model Mice

Intraperitoneal injection of the combination of all three drugs alleviated mechanical allodynia in SNL model mice in a dose-dependent manner (Interaction: DF = 56, F (56,344) = 3.982, *p* < 0.0001; Row Factor: DF = 7, F (7,344) = 110.4, *p* < 0.0001; Column Factor: DF = 8, F (8,344) = 35.68, *p* < 0.0001; two-way ANOVA followed by Bonferroni’s post hoc test). Pre10 + Dul10 + Tra10 injection had the most significant effect in ameliorating mechanical allodynia, and its analgesic effect lasted the longest. The anti-allodynic effect of this drug combination was observed within 30 min after injection, peaked at 90 min, and lasted for 180 min (*p* < 0.005, *p* < 0.0001, *p* < 0.0001, *p* < 0.0001, *p* < 0.0001, and *p* < 0.0001 at 30, 60, 120, and 180 min). The Pre15 + Dul15, Dul30 + Tra10 group exhibited obvious relief of allodynia at 60, 90, and 120 min (*p* < 0.001, *p* < 0.0001, and *p* < 0.001 at 60, 90, and 120 min, respectively). Pre20 + Tra10 injection had obvious anti-allodynic effects at 90 (*p* < 0.0001) and 120 min (*p* < 0.05), while Dul10 + Tra5 and Pre5 + Dul5 + Tra5 injection exerted a significant analgesic effect at 90 min (*p* < 0.005). Pre10 + Tra5 and Pre7.5 + Dul7.5 were not found to have a significant effect on allodynia at any time point (Figure 2B).

### 3.4. Drug Combinations Reduce GFAP Expression in CPIP and SNL Model Mice

In CPIP model mice, Pre10 + Dul10 + Tra10 injection significantly reduced the expression of GFAP in the spinal cord (Figure 3E), as the optical density of the Pre10 + Dul10 + Tra10 group was lower than that of the normal saline group (Figure 3F) (*p* < 0.0001) (Treatment [between columns]: DF = 5, F (5,26) = 38.63, *p* < 0.0001; one-way ANOVA followed by Bonferroni’s post hoc test). The expression of GFAP in the DRG was also significantly reduced in CPIP model mice treated with Pre10 + Dul10 + Tra10 (Figure 3K) compared with CPIP model mice in the normal saline group (*p* = 0.0004) (Treatment [between columns]: DF = 5, F (5,26) = 7.229, *p* = 0.0002; one-way ANOVA followed by Bonferroni’s post hoc test) (Figure 3L). The optical density in the spinal cord was significantly in CPIP model mice in the Pre20 + Tra10, Pre15 + Tra15 and Dul30 + Tra10 groups than in those in the normal saline group (Figure 3B–D,F) (*p* < 0.0001). The expression of GFAP in the DRG was reduced in CPIP model mice in the Pre20 + Tra10 group compared with those in the normal saline group (*p* = 0.0321) (Figure 3H,L). The optical density of the Pre15 + Tra15 and Dul30 + Tra10 group was significantly lower than that of the normal saline group (*p* = 0.0076 and *p* = 0.0051) (Figure 3I,J,L). There was no difference in optical density between the sham operation group (Figure 3A,G) and the Pre10 + Dul10 + Tra10 group (*p* = 0.875 and *p* = 0.2186 for the spinal cord and DRGs, respectively). However, the optical density was significantly different between the sham operation group and the normal saline group (*p* < 0.0001 for both the spinal cord and DRGs) (Figure 4).

In SNL model mice, Pre10 + Dul10 + Tra10 injection significantly reduced the expression of GFAP in the spinal cord (Figure 5E) (Treatment [between columns]: DF = 5, F (5,26) = 5.142, *p* = 0.0021) and DRGs (Treatment [between columns]: DF = 5, F (5,26) = 10.97, *p* < 0.0001; two-way ANOVA followed by Bonferroni’s post hoc test) (Figure 5K). The optical density in the Pre10 + Dul10 + Tra10 group was lower than that in the normal saline group (Figure 5F,L) (*p* = 0.0016 and *p* < 0.0001 for the spinal cord and DRGs, respectively). The expression of GFAP in the spinal cord was reduced in SNL model mice in the Pre20 + Tra10 group compared with those in the normal saline injection group (*p* = 0.0361) (Figure 5B,F). GFAP expression in the DRGs was significantly reduced in the normal saline group (*p* = 0.0003) (Figure 5H,L). The optical density in the spinal cord was not significantly different between SNL model mice in the Pre15 + Tra15 and Dul30 + Tra10 group and those in the normal saline group (Figure 5C,D,F) (*p* = 0.0669 and *p* = 0.163). There was a significant difference in optical density in the DRGs between the Pre15 + Tra15 and Dul30 + Tra10 group and the normal saline injection group (Figure 5I,J,L) (*p* = 0.0015 and *p* = 0.0069). However, there was no significant difference in optical density between the normal saline group and sham operation group (*p* = 0.0037 and *p* < 0.0001 for the spinal cord and DRGs, respectively) (Figure 6).

### 3.5. Rotarod Test

The experimental mice did not experience any side effects, such as dizzeness or drowsiness, and no motor weakness was observed on the rotarod test in the CPIP (Interaction: DF = 28, F (28,184) = 0.9306, *p* = 0.5703; Row Factor: DF = 7, F (7,184) = 11.42, *p* < 0.0001; Column Factor: DF = 4, F (4,184) = 3.130, *p* = 0.0161; two-way ANOVA followed by Bonferroni’s post hoc test) and SNL (Interaction: DF = 28, F (28,184) = 0.8954, *p* = 0.6209; Row Factor: DF = 7, F (7,184) = 3.948, *p* = 0.0005; Column Factor: DF = 4, F (4,184) = 2.966, *p* = 0.0209; two-way ANOVA followed by Bonferroni’s post hoc test) model (*p* > 0.9999) (Figure 7).

## 4. Discussion

Our research results show that the CPIP and SNL models exhibited mechanical allodynia and that the combination of pregabalin, duloxetine, and tramadol can alleviate pain in these models better than any combination of two of these drugs. The expression of GFAP was decreased after treatment with the drug combination, and histology confirmed that the expression of GFAP was increased in the two different pain models.

Post-traumatic inflammation is a key mechanism underlying the symptoms of CPIP model mice, which exhibit clinical symptoms of complex regional pain syndrome. Additionally, during NP, the levels of inflammatory cytokines increase, leading to the activation of glial cells, especially microglia and astrocytes in the spinal cord and brain, which play a prominent role in pain [15,16,17,18]. After astrocyte activation, the expression of GFAP is increased, and then activated astrocytes release inflammatory stimuli, such as cytokines and neurotrophic factors, which leads to central sensitization by altering the polarization properties of afferent neurons and changes in the development and maintenance of the pain transmission pathway [19].

The dorsal root ganglion (DRG) is a site in the peripheral somatosensory pathway that is closely related to the development and maintenance of chronic pain [20]. A unique feature of the dorsal root ganglia is that each sensory neuron cell body and its initial axonal segment are encased by several satellite glial cells (SGCs) that are interconnected by gap junctions that together form an independent anatomical and functional sensory unit [21,22]. There is now increasing evidence that normal neuronal activity in the DRG relies on neuronal–glial interactions and that SGC dysfunction may contribute to painful conditions [23,24,25]. The current study demonstrated that GFAP expression was used to demonstrate SGC activation in various models of traumatic nerve injury [26] as well as in models of type 1 diabetes [27], ischemia [28], facial cancer [29], chemotherapy administration [30,31], and peripheral inflammation [32]. Although far from exhaustive, this series of studies demonstrates a strong correlation between the enhancement of GFAP immunoreactivity and the induction of astrocytes and SGC reactivity in mouse experimental models.

Protein kinase C epsilon (PKCε) and TRPV1 are expressed in the axons and terminals of astrocytes in layers I-II of the spinal dorsal horn [33,34]. However, whether there is a close connection between astrocytes and the PKCε/TRPV1 signaling pathway is still unclear. Pregabalin can inhibit the activity of PKCε to a certain extent, thereby reducing the sensitivity of TRPV1, inhibiting inflammation, significantly reducing the activation of astrocytes, and alleviating NP [35]. Alterations that occur in astrocytes include the release of different cytokines and overexpression of GFAP, which is a common astrocyte marker that plays an important role in the development of hyperalgesia and chronic pain [36].

Duloxetine protects the PNS and central nervous system during NP, at least in part by decreasing the number of spinal astrocytes and microglia [37]. This is because duloxetine and other similar serotonin and norepinephrine reuptake inhibitors (SNRIs) can regulate neuroinflammation by interacting with serotonin and norepinephrine receptors on microglia [38].

Tramadol’s analgesic effect against NP is believed to involve at least in part its interaction with presynaptic and postsynaptic α2-ARs [39,40,41]. The analgesic effects of tramadol and α2-AR agonists are enhanced in NP, possibly due to the increased activity of presynaptic and postsynaptic α2-ARs [42,43,44]. In addition, dexmedetomidine is a highly selective α2-AR agonist that inhibits the activation of spinal astrocytes and extracellular signal-regulated kinase (ERK) signaling, possibly through the activation of astrocytes. α2-AR expressed on cells inhibits inflammation and NP [45,46]. Tramadol can exert analgesic effects through μ-opioid receptors and prevent and relieve NP through α2-ARs, with the latter effect being due to α2–AR-mediated inhibition of astrocyte activation [47]. These results suggesting that pregabalin, duloxetine, and tramadol inhibit the activation of astrocytes or microglia through their respective mechanisms.

In this study, the combination of the three drugs alleviated mechanical allodynia better than any of the two drug combinations in both CPIP and SNL model mice. Additionally, it altered GFAP expression in the spinal cord and DRGs. Moreover, the combination of the three drugs did not cause any side effects or motor dysfunction in the mice. However, this study has a limitation. We only performed immunohistochemistry and did not evaluate protein expression.

## 5. Conclusions

In conclusion, the combination of pregabalin, duloxetine, and tramadol produced an anti-allodynic effect in CPIP and SNL model mice, as indicated by a behavioral test, and inhibited astrocytes, as shown by immunohistochemistry. The enhanced efficacy and decreased side effects such as dizziness, drowsiness, and motor weakness for these combinations will aid physicians in clinical treatment. Further investigations are needed in the clinical setting.

## Figures and Tables

**Figure 1 pharmaceutics-14-00670-f001:**
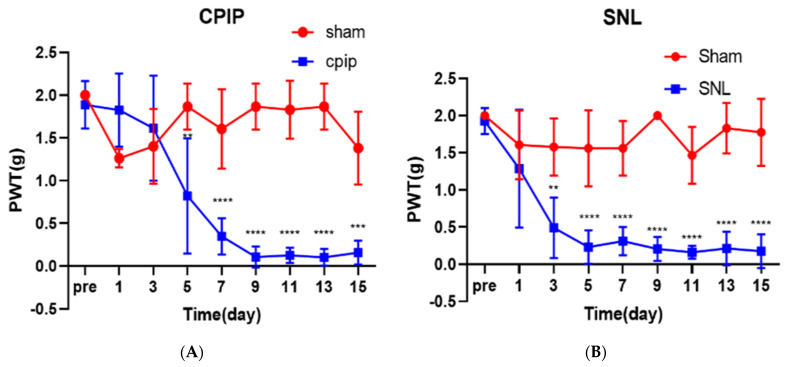
The Von Frey test showed the time course of allodynia in the left hind paw of CPIP (*n* = 6) mice and Sham (*n* = 4) group, SNL (*n* = 6) mice and Sham (*n* = 4) group mice. (**A**) During the entire 15-day test, the withdrawal threshold of the mice in the Sham group did not change significantly. The paw withdrawal threshold of the CPIP mouse was significantly decreased on the 4th day after reperfusion and continued to the 15th day. (**B**) During the 15-day test, the withdrawal threshold of the mice in the Sham group did not change significantly. The withdrawal threshold of SNL mice was significantly reduced and lasted for 15 days after reperfusion. CPIP, chronic post-ischemic pain; SNL, spinal nerve ligation. ** *p* < 0.005, *** *p* < 0.001, **** *p* < 0.0001 at each time point compared to that in the normal saline.

**Figure 2 pharmaceutics-14-00670-f002:**
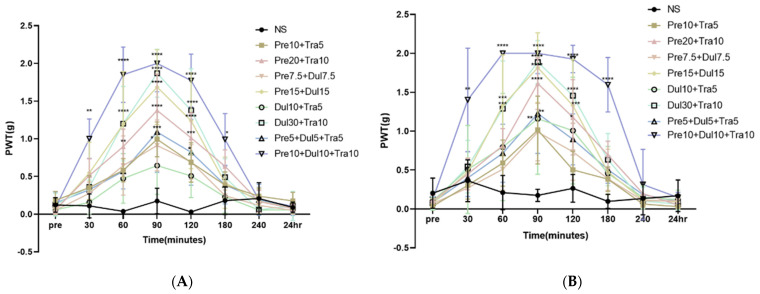
The effect of mixed administration of pregabalin + duloxetine + tramadol on the tactile threshold of CPIP and SNL mice. (**A**) Compared with the control group, drug injection can reduce the mechanical allodynia of CPIP mice in a dose-dependent manner. pregabalin10 mg/kg + duloxetin10 mg/kg + tramadol10 mg/kg injection group has the best relieving effect on mechanical allodynia. (**B**) The mechanical allodynia of the mice in each group injected with the drug was significantly reduced. pregabalin10 mg/kg + duloxetine10 mg/kg + tramadol10 mg/kg group has the strongest relieving effect on mechanical allodynia and continued effective (*n* = 4 in NS group, *n* = 6 in other treatment groups). NS, normal saline; Pre, pregabalin; Dul, duloxetine; Tra, Tramadol; CPIP, chronic post-ischemic pain; SNL, spinal nerve ligation. * *p* < 0.05, ** *p* < 0.005, *** *p* < 0.001, **** *p* < 0.0001 at each time point compared to that in the normal saline.

**Figure 3 pharmaceutics-14-00670-f003:**
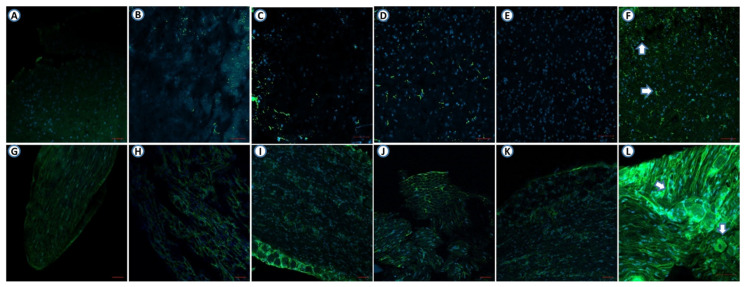
The effect of intraperitoneal injection of mixed drugs on the expression of GFAP (luminous green) in the spinal cord and dorsal root ganglion of the CPIP model. Original magnification; ×200. (**A**) GFAP immunohistochemical staining of the spinal cord of mice in sham operation group *(n* = 4). (**B**) Pre20 mg/kg + Tra10 mg/kg-injected CPIP mouse (*n* = 6). (**C**) Pre15 mg/kg + Dul15 mg/kg-injected CPIP mouse (*n* = 6). (**D**) Dul30 mg/kg + Tra10 mg/kg-injected CPIP mouse (*n* = 6). (**E**) Pre10 mg/kg + Dul10 mg/kg + Tra10 mg/kg-injected CPIP mouse (*n* = 6). (**F**) normal saline-injected CPIP mouse (*n* = 4). (**G**) GFAP immunohistochemical staining of the DRG of mice in sham operation group (*n* = 4). (**H**) Pre20 mg/kg + Tra10 mg/kg-injected CPIP mouse (*n* = 6). (**I**) Pre15 mg/kg + Dul15 mg/kg-injected CPIP mouse (n = 6). (**J**) Dul30 mg/kg + Tra10 mg/kg-injected CPIP mouse (*n* = 6). (**K**) Pre10 mg/kg + Dul10 mg/kg + Tra10 mg/kg-injected CPIP mouse (*n* = 6). (**L**) normal saline-injected CPIP mouse (*n* = 4). GFAP, glial fibrillary acidic protein; DRG, dorsal root ganglia; CPIP, chronic post-ischemic pain; Pre, pregabalin; Dul, duloxetine; Tra, tramadol.

**Figure 4 pharmaceutics-14-00670-f004:**
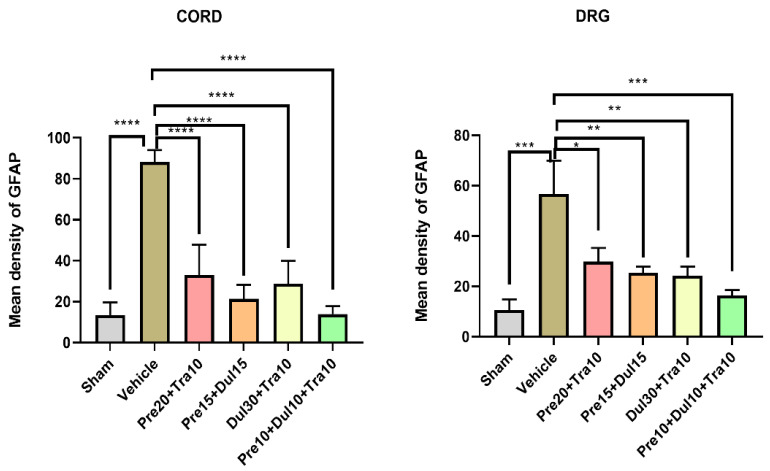
Histogram representing the optical density of GFAP in spinal cord, DRG from sham *(n* = 4), and normal saline-treated (*n* = 4) CPIP mice. Pre20 mg/kg + Tra10 mg/kg-treated (*n* = 6) CPIP mice, Pre15 mg/kg + Dul15 mg/kg-treated (*n* = 6) CPIP mice, Dul30 mg/kg + Tra10 mg/kg-treated (*n* = 6) CPIP mice, Pre10 mg/kg + Dul10 mg/kg + Tra10 mg/kg-treated (*n* = 6) CPIP mice. The expression of GFAP in the four treated group was significantly decreased compared to the normal saline treated group. GFAP, glial fibrillary acidic protein; DRG, dorsal root ganglia; Pre, pregabalin; Dul, duloxetine; Tra, tramadol; CPIP, chronic post-ischemic pain. * *p* < 0.05, ** *p* < 0.005, *** *p* < 0.001, **** *p* < 0.0001.

**Figure 5 pharmaceutics-14-00670-f005:**
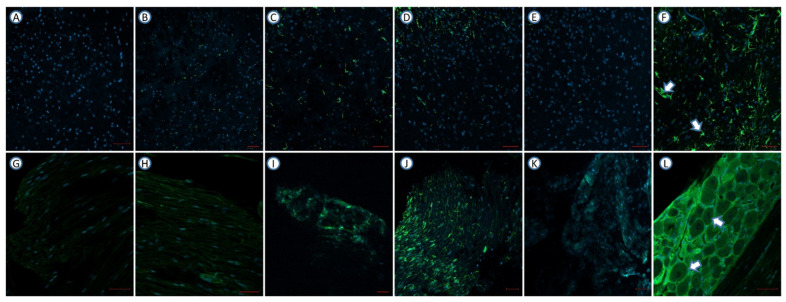
The effect of intraperitoneal injection of mixed drugs on the expression of GFAP (luminous green) in the spinal cord and dorsal root ganglion of the SNL model. Original magnification; ×200. (**A**) GFAP immunohistochemical staining of the spinal cord of mice in sham operation group (*n* = 4). (**B**) Pre20 mg/kg + Tra10 mg/kg-injected SNL mouse (*n* = 6). (**C**) Pre15 mg/kg + Dul15 mg/kg-injected SNL mouse (*n* = 6). (**D**) Dul30 mg/kg + Tra10 mg/kg-injected SNL mouse (*n* = 6). (**E**) Pre10 mg/kg + Dul10 mg/kg + Tra10 mg/kg-injected SNL mouse (*n* = 6). (**F**) normal saline-injected SNL mouse (*n* = 4). (**G**) GFAP immunohistochemical staining of the DRG of mice in sham operation group (*n* = 4). (**H**) Pre20 mg/kg + Tra10 mg/kg-injected SNL mouse (*n* = 6). (**I**) Pre15 mg/kg + Dul15 mg/kg-injected SNL mouse (*n* = 6). (**J**) Dul30 mg/kg + Tra10 mg/kg-injected SNL mouse (*n* = 6). (**K**) Pre10 mg/kg + Dul10 mg + Tra10 mg/kg-injected SNL mouse (*n* = 6). (**L**) normal saline-injected SNL mouse (*n* = 4). GFAP, glial fibrillary acidic protein; DRG, dorsal root ganglia; SNL, spinal nerve ligation; Pre, pregabalin; Dul, duloxetine; Tra, tramadol.

**Figure 6 pharmaceutics-14-00670-f006:**
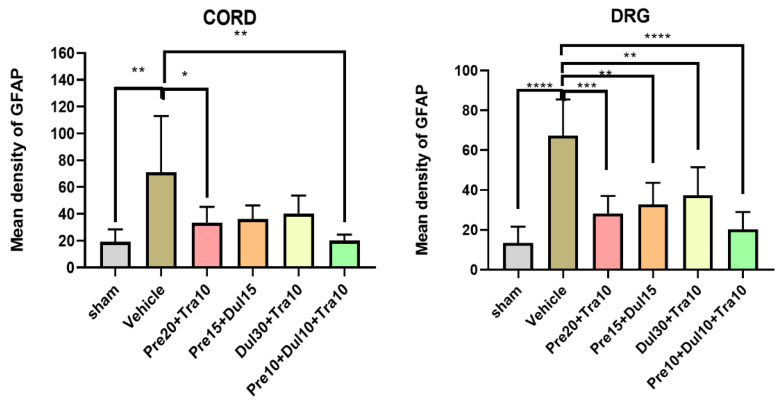
Histogram representing the optical density of GFAP in spinal cord, DRG from sham (n = 4), and normal saline-treated (*n* = 4) SNL mice. Pre20 mg/kg + Tra10 mg/kg-treated (*n* = 6) SNL mice, Pre15 mg/kg + Dul15 mg/kg-treated (*n* = 6) SNL mice, Dul 30 mg/kg + Tra10 mg/kg-treated (*n* = 6) SNL mice, Pre10 mg/kg + Dul10 mg/kg + Tra10 mg/kg-treated (*n* = 6) SNL mice. The expression of GFAP in the four treated group was significantly decreased compared to the normal saline treated group in DRG. GFAP, glial fibrillary acidic protein; DRG, dorsal root ganglia; Pre, Pregabalin; Dul, Duloxetine; Tra, Tramadol; SNL, spinal nerve ligation. * *p* < 0.05, ** *p* < 0.005, *** *p* < 0.001, **** *p* < 0.0001.

**Figure 7 pharmaceutics-14-00670-f007:**
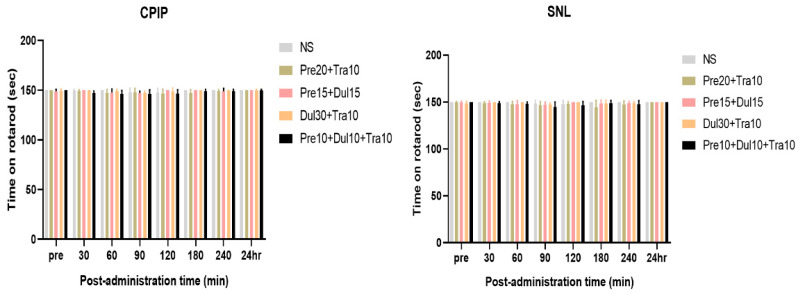
The rotarod time of mixed drugs-injected mice. Rotarod time was observed before (Pre) and after mixed drugs or saline injection. Rotarod time was not decreased by injection of normal saline (NS) and mixed drugs. (*n* = 4 in saline group, *n* = 6 other groups) (*p* > 0.9999 vs. NS).

## Data Availability

The data presented in this study are available on request from the corresponding author.

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
