# Peer review of "Effect of Pregabalin Combined with Duloxetine and Tramadol on Allodynia in Chronic Postischemic Pain and Spinal Nerve Ligation Mouse Models"

_pharmaceutics, 2022, doi:10.3390/pharmaceutics14030670_

Round 1
Reviewer 1 Report
In this study the authors test the effect of combination of three analgesics commonly used in clinic; pregabalin, tramadol and duloxetine in two mouse models of neuropathic pain. The authors found that combination of three drugs at low dose alleviates mechanical allodynia induced by chronic postischemic pain and by spinal nerve ligation. Treatment seems to prevent the overexpression of GFAP in spinal cord and DRG induced observed in both models. Authors suggest combining these drugs in clinic may enhanced their efficacy and decrease side effect in patients experiment neuropathic pain. However, there are a number of weaknesses in the work that reduce enthusiasm.
I have a few comments/questions.
- At what point after surgery were the treatments administered? Justify.
- Only male mice were used- why were females excluded.
- The doses were not justified. Idem, there miss a link with the clinic.
- In many figure legends the n values are missing.
- Were the studies carried out in blinded fashion and how was blinding done?
- Why the rotarod test data are presented? At the end of the discussion section, “the combination of the three drugs did not cause any side effects or motor dysfunction”. It necessary to detailed and show these results (and Method).
- Representatives examples of IHC study are not convincing and of poor quality. It difficult to distingue a DRG section on Figure 3a, 3b and 3d, for example. Figure 3F and 3f are saturated, as show by the DAPI staining which is very clear compared with other images.
- Ref 1 do not appropriate.
- Line 56-57: need a reference
- It is surprising that none antalgic (ketamine, xylocaine) was used during the surgeries
- Line 96: “from 2.44 to 4.31 g” have to be changed by “0.04 to 2 g”
- “type of response”, could you detail?
- Line 127: PB need to be defined
- The name of the vehicle group need to be homogenize in the text; normal saline, or vehicle. Line 203: “NS: not significant”è normal saline?
- The discussion is focused on GFAP and astrocytes, but there is no astrocyte in the peripheral nervous system (DRG). Glial cells of DRG are the satellite glial cells, this must be mentioned.
Author Response
Dear Editor:
Thank you for providing these insights.
It is our privilege to work with Pharmaceutics and the associated editor and reviewers.
We have modified the manuscript based on your perspectives.
In the responses, we have denoted the changes made by using red colored font in the manuscript. Also, we have written our responses in blue colored font in our “Response to Reviewer” document.
We look forward to further opportunity to work with your journal in the future.
Sincerely,
Dr. Hue Jung Park.
Department of Anesthesia and Pain Medicine, Seoul St Mary's Hospital, College of Medicine, The Catholic University of Korea E-mail: huejung@catholic.ac.kr
Mobile: 82-10-3226-2047

Reviewer 2 Report
The paper presents interesting data on the effect induced by various combinations of drugs potent in the treatment of neuropathic pain. Nevertheless, the manuscript needs some corrections and improvement before its publishing.
1. The Authors used 5 mice per group. Please provide whether any power analysis was done.
2. Please provide in the methodology the total number of mice used.
3. Drug administration subsection should be placed before the induction of mechanical allodynia.
4. I'm not sure on what basis did the Authors choose the indicated doses of each drugs and their combinations. In line to this, please provide the ED50 for each drug alone. In my opinion, the Authors should add the results for the drug administered solely in this specific experimental conditions. This would improve greatly the manuscript as it will give the possibility to analyze the character of interactions between both drugs used (i.e., synergistic, additive, inhibitory).
5. Every drug was administered with another at a specific dose. Nonetheless, I'm wondering whether were the drugs administered simultaneously? If the answer is positive, then please provide the way (a mixture?, both-sided intraperitoneal injection?).
6. The Rotarod test should be described as a subsection not introduced in the subsection 2.5.
7. Did the Authors use the same group of animals in order to analyze the expression of GFAP? I assume that these were additional groups since the animals were sacrificed 90 min after drug administration. Therefore, please provide on what basis did the Authors choose the time point of 90 min?
8. In the introduction section, the Authors indicated that the paper aimed at evaluating effective drug combination with a desired safety profile. Unfortunately, there is no deep analysis on possible side effects induced by the above-mentioned drug combination. Rotarod is not enough, especially when consider tramadol! Therefore, the Authors should either modify the last sentence in the introduction or provide sufficient results revealing drugs safety profile.
9. Sham group was subjected to the operation without the ligation of the spinal nerve, and thereafter was injected with saline as a control group. However, in order to provide relevant results, the Authors should have provided also with a control with SNL and saline.
10. Results: Each analysis needs to be verified, and n's provided (in addition to the F parameter, df, and p value).
11. The conclusions aren't supported with the results, especially when given side effects of the drug combination. The Authors did not performed any study for this (tolerance, open-field test, drug impact on intestinal mobility, etc.)
Author Response

(The authors gave the same response as above.)

Round 2
Reviewer 2 Report
the paper is now acceptable to be published